# Type I interferon underlies severe disease associated with Junín virus infection in mice

Brady T Hickerson[1†], Eric J Sefing[1], Kevin W Bailey[1], Arnaud J Van Wettere[1,2], Manuel L Penichet[3,4,5,6,7], Brian B Gowen[1*]

[1]Department of Animal, Dairy and Veterinary Sciences, Utah State University, Logan, United States; [2]Utah Veterinary Diagnostic Laboratory, Utah State University, Logan, United States; [3]Division of Surgical Oncology, Department of Surgery, David Geffen School of Medicine at University of California, Los Angeles (UCLA), Los Angeles, United States; [4]Department of Microbiology, Immunology and Molecular Genetics, David Geffen School of Medicine at UCLA, Los Angeles, United States; [5]UCLA Molecular Biology Institute, Los Angeles, United States; [6]UCLA Jonsson Comprehensive Cancer Center, Los Angeles, United States; [7]UCLA AIDS Institute, Los Angeles, United States

*For correspondence:
brian.gowen@usu.edu

Present address: †Division of Biotechnology Review and Research-III, Office of Biotechnology Products, Center for Drug Evaluation and Research, Food and Drug Administration, Silver Spring, United States

**Abstract** Junín virus (JUNV) is one of five New World mammarenaviruses (NWMs) that causes fatal hemorrhagic disease in humans and is the etiological agent of Argentine hemorrhagic fever (AHF). The pathogenesis underlying AHF is poorly understood; however, a prolonged, elevated interferon-α (IFN-α) response is associated with a negative disease outcome. A feature of all NWMs that cause viral hemorrhagic fever is the use of human transferrin receptor 1 (hTfR1) for cellular entry. Here, we show that mice expressing hTfR1 develop a lethal disease course marked by an increase in serum IFN-α concentration when challenged with JUNV. Further, we provide evidence that the type I IFN response is central to the development of severe JUNV disease in hTfR1 mice. Our findings identify hTfR1-mediated entry and the type I IFN response as key factors in the pathogenesis of JUNV infection in mice.

## Introduction

The etiological agents of the complex of South American arenaviral hemorrhagic fevers are a group of closely related New World mammarenaviruses (NWMs) including Junín, Machupo, Guanarito, Sabiá and Chapare. These viruses are maintained in nature through persistent infections of their respective host rodent species (*Salazar-Bravo et al., 2002*). Transmission of these agents to humans occurs through inhalation of aerosolized virus particles or contact with virus-containing rodent secreta or excreta (*Childs et al., 1995*). Junín virus (JUNV) is the most prevalent of pathogenic NWMs and is the etiological agent of Argentine hemorrhagic fever (AHF). AHF is an insidious disease characterized by severe hemorrhagic and/or neurologic manifestations with remittent fever, tremors, vascular leak and shock (*de Bracco et al., 1978*; *Elsner et al., 1973*; *Enría et al., 1986*; *Enria et al., 2008*). The pathogenesis of AHF is poorly understood; however, a salient feature of severe disease is an elevation in serum interferon-α (IFN-α) and other inflammatory mediators (*Marta et al., 1999*; *Levis et al., 1984*; *Levis et al., 1985*; *Heller et al., 1992*). The case-fatality rate of AHF can be as high as 30% in untreated individuals and the only countermeasures available for the prevention and treatment of severe JUNV infections are the Candid #1 vaccine and convalescent plasma (*Enria et al., 2008*). Neither has been approved for use outside of the area of endemicity and recent studies suggest that reversion to virulence by the vaccine virus can occur through a single nucleotide

change in the envelope glycoprotein gene (*York and Nunberg, 2018*; *Seregin et al., 2015*; *Albariño et al., 2011*).

Animal models are essential to gaining insights into viral pathogenesis. Several species of non-human primates are susceptible to lethal JUNV disease (*Kenyon et al., 1992*; *McKee et al., 1985*; *Avila et al., 1987*). These models are considered the 'gold standard' because JUNV infection in these species more closely recapitulates the human disease. However, these models are costly and require specialized primate housing facilities within maximum biocontainment. Few small-animal models for JUNV exist because most standard laboratory rodent species, such as mice and hamsters, are refractory to severe disease. The most commonly used rodent model for JUNV is the guinea pig, which has served as the primary animal model to investigate pathogenesis and for screening promising therapeutic interventions (*Yun et al., 2008*; *Gowen et al., 2013*; *Zeitlin et al., 2016*). However, the use of this model for investigations into JUNV pathogenesis and countermeasure development has been mired by the lack of commercial reagents for this species.

A feature specific to the pathogenic NWMs is the use of human transferrin receptor 1 (hTfR1) for cellular entry (*Radoshitzky et al., 2007*; *Helguera et al., 2012*). In addition, different species known to be susceptible to disease following challenge with pathogenic NWMs express TfR1 orthologs that bind the viral envelope glycoprotein facilitating entry (*Helguera et al., 2012*; *Hickerson et al., 2020*). These findings suggest that the use of TfR1 for viral attachment and entry is an important determinant in defining whether a NWM can cause severe disease in a species other than the respective rodent reservoir host. Based on this, we investigated whether the expression of hTfR1 in laboratory mice would render them susceptible to lethal disease following JUNV challenge. Here, we demonstrate that transgenic hTfR1 mice develop a lethal disease course when exposed to the pathogenic Romero strain of JUNV and characterize the natural history and pathogenesis of disease. We also show that the type I IFN response plays a central role in the development of severe JUNV infection and disease in mice expressing hTfR1. The development of the hTfR1 mouse model of JUNV infection provides a novel system to investigate viral pathogenesis and assess promising therapeutics.

## Results

### Susceptibility of hTfR1 mice to JUNV infection

Wild-type (WT) mice are refractory to disease following JUNV challenge (*Golden et al., 2015*). To investigate whether expression of hTfR1 would confer susceptibility to disease following JUNV infection, groups of 3-week-old WT, hTfR1 heterozygous (HET) and hTfR1 homozygous (HOM) mice were challenged by intraperitoneal (i.p.) injection with a $10^5$ fifty percent cell culture infectious dose (CCID$_{50}$) of JUNV, or sham-infected, and observed for mortality and weight change (*Figure 1A and B*). On day 8 post-infection (p.i.), initial signs of lethargy and ruffling of fur accompanied weight loss in the cohort of hTfR1 HOM mice, which increased in severity until they succumbed to infection by day 16 p.i. JUNV disease was also observed in the hTfR1 HET mice but to a lesser degree with the mice exhibiting stagnated weight gain beginning on day 9 p.i with a single animal requiring euthanasia due to severe neurologic signs on day 14 p.i. In contrast, the WT mice showed no visibly apparent signs of illness, suggesting that expression of hTfR1 was necessary for the development of JUNV disease.

### Age- and dose-dependent susceptibility of hTfR1 mice to lethal JUNV infection

To gain further insight into the susceptibility of hTfR1 HOM mice to lethal JUNV infection, groups of 3, 4, 5 and 6-week-old animals were challenged i.p. with $10^5$ CCID$_{50}$ of JUNV and observed for morbidity and mortality. The mice were also scored for clinical disease signalment to further characterize the disease course and presentation. As shown in *Figure 2*, susceptibility to JUNV disease decreased with age. Beginning one-week p.i., the 3-week-old mice started to display clinical disease signs accompanied by weight loss and uniform lethality by day 13 p.i. (*Figure 2A,B and D*). In the 4-week-old animals, clinical disease signs and stagnation in weight gain started to develop by day 8 p.i., with one animal succumbing to infection 13 days after JUNV challenge (*Figure 2A,B and D*). No mortality was observed in the 5-week-old animals despite notable weight loss and other signs of

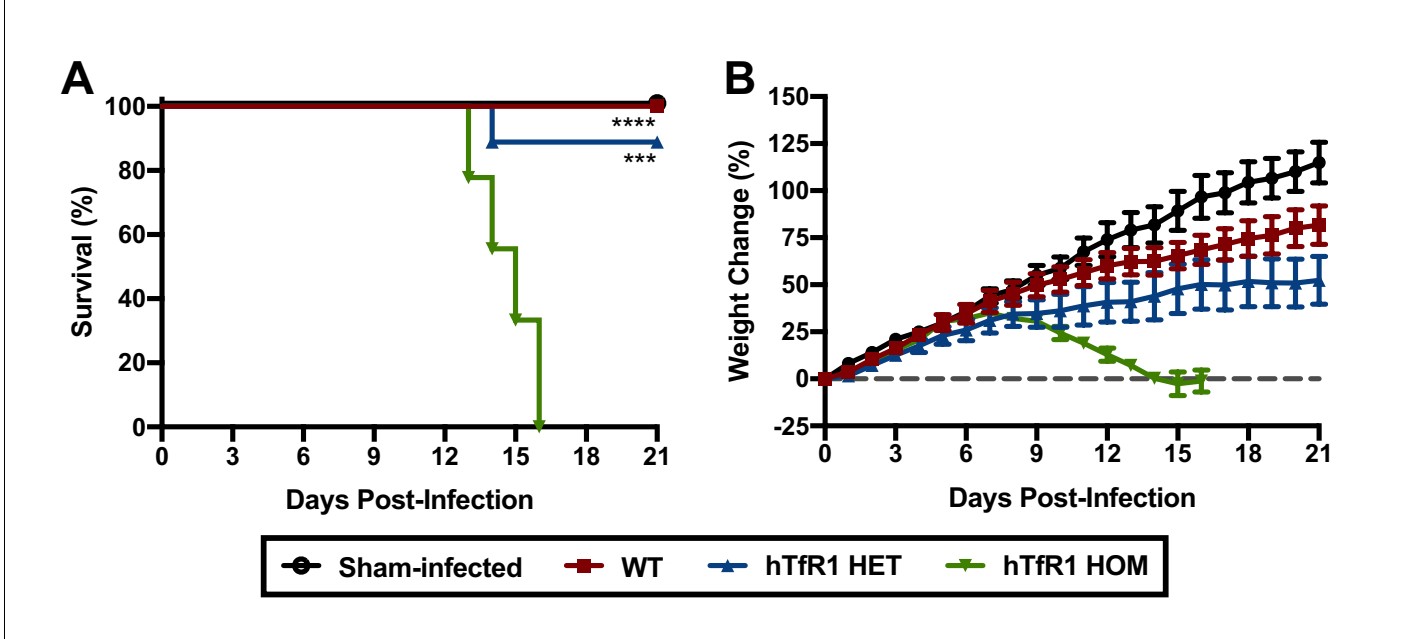

**Figure 1.** Expression of hTfR1 results in severe disease following JUNV infection. Three-week-old WT, hTfR1 HET or hTfR1 HOM mice ($n$ = 8–9/group) were infected i.p. with $10^5$ $CCID_{50}$ of JUNV and monitored daily. (**A**) Survival and (**B**) weight change of animals relative to the day of virus challenge (group mean and standard error of the mean; SEM) are shown. Sham-infected controls ($n$ = 6) consisted of a mix of WT, hTfR1 HET and hTfR1 HOM hTfR1 mice. Aggregate data from two independent experiments. ***p=0.0003 and ****p<0.0001 compared to hTfR1 HOM mice.

The online version of this article includes the following source data for figure 1:

**Source data 1.**

disease starting on day 8 p.i. (***Figure 2A,C and D***). There was no evidence of weight loss or observable clinical signs in the 6-week-old mice (***Figure 2C and D***).

Based on their susceptibility to lethal disease following JUNV challenge, we sought to determine the 50% and 90% lethal challenge doses ($LD_{50}$ and $LD_{90}$, respectively) in 3-week-old hTfR1 HOM mice. Groups of mice were challenged i.p. with serial $\log_{10}$ dilutions of JUNV ($10^5$, $10^4$ and $10^3$ $CCID_{50}$) or sham-infected and observed for weight loss, other clinical disease signs and mortality. A JUNV challenge of $10^5$ or $10^4$ $CCID_{50}$ resulted in uniform lethality by 15 and 18 days p.i., respectively, while 2 of 7 animals survived a challenge dose of $10^3$ $CCID_{50}$ (***Figure 3A***). The onset of weight loss and other clinical disease signs also occurred in a dose-dependent manner, with animals challenged with higher doses of virus developing disease earlier in the course of infection (***Figure 3B and C***). All animals inoculated with a challenge dose of $10^5$ $CCID_{50}$ started losing weight and exhibiting other clinical disease signs such as ruffled fur and lethargy by day 8 p.i. (***Figure 3B and C***). A one- to two-day delay in weight loss and other disease signs was observed in the groups challenged with $10^4$ and $10^3$ $CCID_{50}$ of JUNV, respectively. Based on the mortality results in 3-week-old hTfR1 HOM mice, the JUNV $LD_{50}$ was less than 1000 $CCID_{50}$ and the $LD_{90}$ was calculated to be 3793 $CCID_{50}$.

## Natural history and pathogenesis of JUNV infection in hTfR1 mice

To gain insight into the pathogenesis of JUNV in the hTfR1 HOM mice, a natural history study was designed to investigate several virologic, clinical and laboratory parameters during the course of infection. To ensure uniform lethality, a challenge dose of $10^4$ $CCID_{50}$ was administered i.p. to 3-week-old animals. As shown in ***Figure 4A***, the JUNV-challenged mice began to plateau in weight gain on day 9 p.i. and started to lose weight by day 10 p.i. Clinical disease signs were observed in a few mice as early as day 8 p.i., which corresponded with a sharp increase in serum IFN-α concentration (***Figure 4B and C***). By day 12 p.i., all mice in the day 12 sacrifice group had succumbed, with only 2 of 4 animals remaining in the group scheduled for sacrifice on day 14 p.i. These 2 mice were moribund and therefore euthanized on day 12 p.i. and assessed for IFN-α levels, viremia and tissue

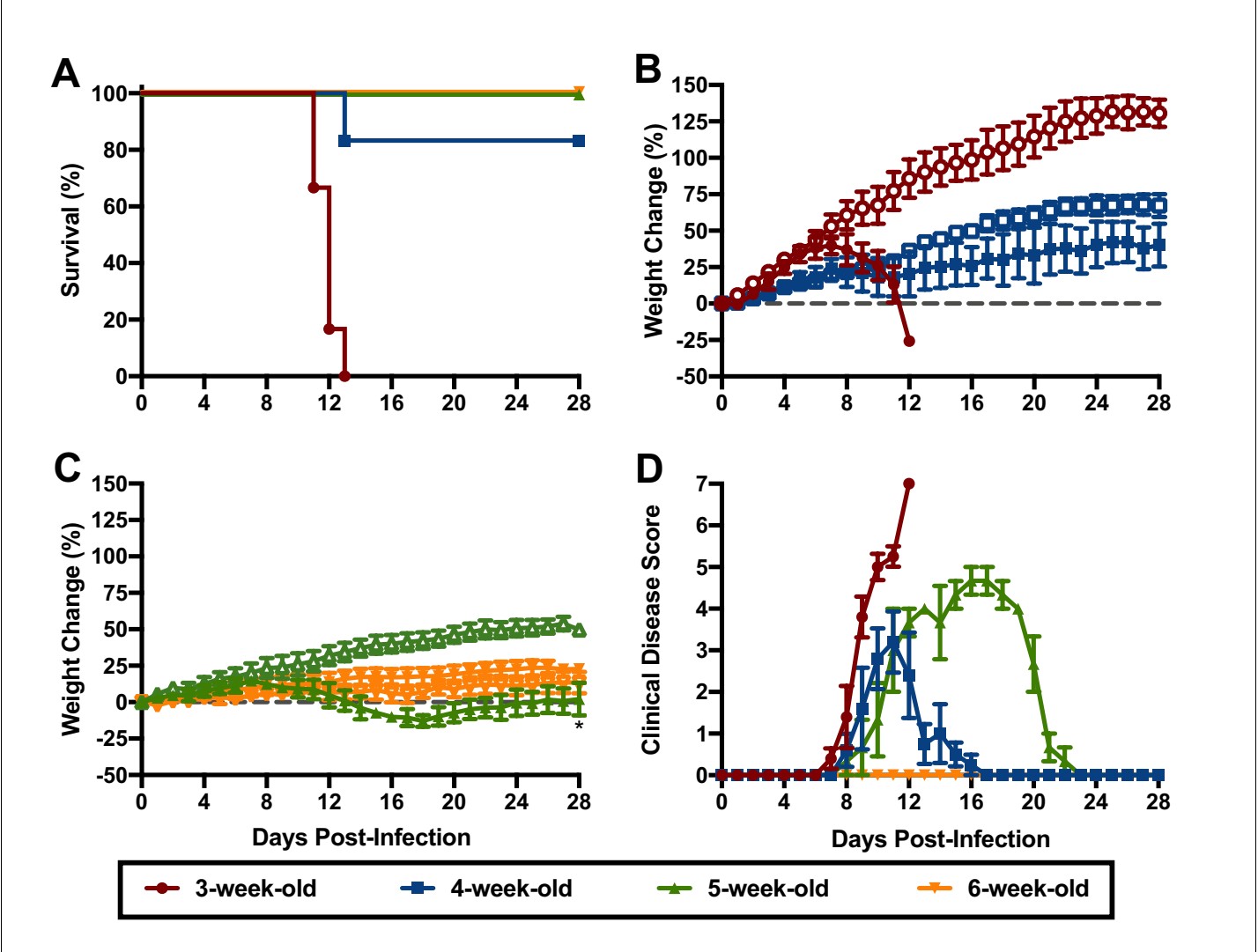

**Figure 2.** Age-dependent susceptibility of hTfR1 HOM mice to lethal JUNV disease. Shown are (**A**) survival (*n* = 6/group for the 3- and 4-week-old mice, *n* = 3/group for the 5- and 6-week-old mice), (**B**) weight change in 3 to 4 week-old mice, (**C**) weight change in 5 to 6 week-old mice and D) clinical disease scores for each group infected i.p. with $10^5$ $CCID_{50}$ of JUNV. Open symbols in the weight change graphs indicate age-matched sham-infected controls (*n* = 3). The weight data are represented as the group mean and SEM of the percent change in weight of animals relative to their starting weights on the day of virus challenge. Clinical scoring is expressed as group mean and SEM. *p=0.0448 compared to age-matched sham-infected controls.

The online version of this article includes the following source data for figure 2:

**Source data 1.**

viral burden. High systemic concentrations of IFN-α were present in the serum of all JUNV-infected animals by day 10 p.i., with peak concentrations observed in the moribund day-12 mice (*Figure 4C*). The virus was first detected in the liver, spleen and brain on day 6 p.i. (*Figure 4D*). Thereafter, viral loads generally continued to increase and were detectable in other organs as the infection progressed. Viremia was undetectable in the mice until day 10 p.i. (*Figure 4D*). The two moribund mice that were euthanized on day 12 p.i. had substantial viral loads in serum and all tissues analyzed.

Microscopic analysis revealed neutrophilic encephalitis and individual cell death (necrosis or apoptosis) in the brain and in the splenic red and white pulps in infected mice as early day 10 p.i. (not shown) with more moderate to severe lesions seen on day 12 p.i. (*Figure 5*). The splenic red and white pulp organization was normal, but mild to moderate individual cell necrosis or apoptosis was present in the white pulp and, to a lesser extent, in the red pulp (*Figure 5F and H*). The dead cells

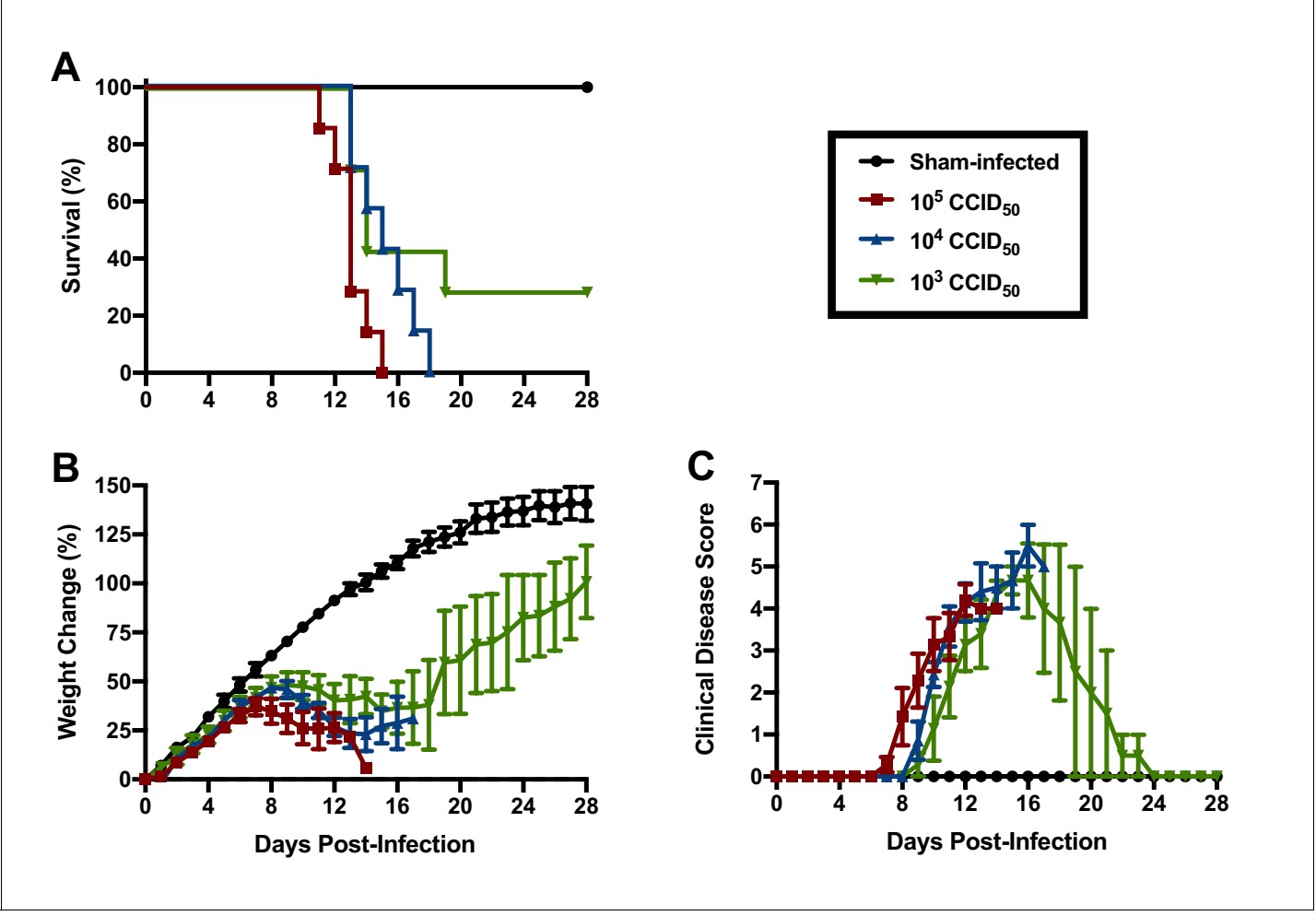

**Figure 3.** JUNV lethal dose determination in 3-week-old hTfR1 HOM mice. Shown are (**A**) survival (*n* = 7/JUNV challenge group, *n* = 3 sham-infected controls), (**B**) weight change relative to the day of virus challenge (group mean and SEM) and (**C**) clinical disease scores (group mean and SEM) for groups of mice infected i.p. with $10^5$, $10^4$ or $10^3$ $CCID_{50}$ of JUNV or sham-infected. $LD_{90}$ = 3793; $LD_{50}$ <1000 $CCID_{50}$. The online version of this article includes the following source data for figure 3:

**Source data 1.**

were presumed to be mainly lymphocytes or macrophages given the location in the white pulp. Immunohistochemistry (IHC) for JUNV antigen was performed on mice euthanized on day 12 p.i. Strong cytoplasmic immunoreactivity was present in neurons multifocally and randomly within the midbrain (*Figure 6B and D*), as well as the cerebral cortex, thalamus and hypothalamus (not shown). Moderate to strong cytoplasmic immunoreactivity was observed in mononuclear cells, mainly within the white pulp of the spleen (*Figure 6F and H*). JUNV antigen was not detected in the kidney, liver, intestine or lung tissue (not shown). The lack of IHC staining for viral antigen in certain tissues with measurable infectious viral loads may be due to a delay in the accumulation of JUNV antigen detectable by IHC, masking of antigen by prolonged formalin fixation of tissues and/or the sensitivity of the IHC staining technique.

## Contribution of the type I IFN response in JUNV pathogenesis in hTfR1 mice

Severe cases of AHF are associated with elevated concentrations of serum IFN-α, which may contribute to disease severity (*Levis et al., 1984*; *Levis et al., 1985*). To investigate whether the type I IFN response contributes to the development of severe disease in hTfR1 HOM mice, we challenged cohorts of animals representing 6 different genetic backgrounds with JUNV. As expected, challenge

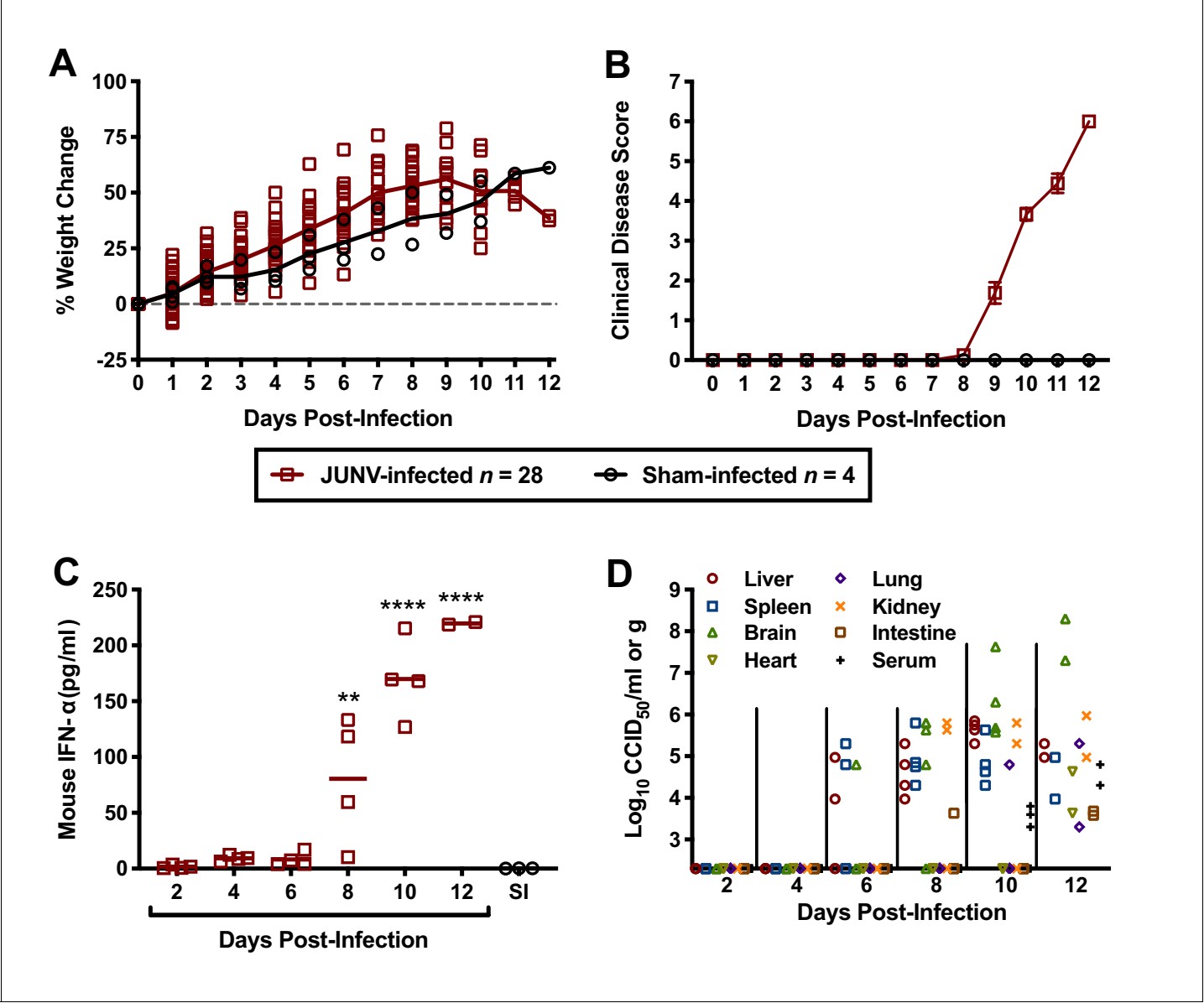

**Figure 4.** Temporal analysis of weight change, clinical disease signs, IFN-α and viral burden during the course of JUNV infection in 3-week-old hTfR1 HOM mice. Animals (n = 28) were infected i.p. with $10^4$ CCID$_{50}$ of JUNV and subsets of 4 mice were designated for euthanasia on days 2, 4, 6, 8, 10, 12 and 14 p.i. for blood and tissue collection and analysis. Due to death prior to sample collection on days 12 and 14, only 2 mice were available on day 12. Four sham-infected animals were included as controls and 3 (one per day) were euthanized on days 2, 6 and 10 for sample collection. (A) Weight change of animals relative to the day of virus challenge, (B) clinical disease scores (mean and SEM), (C) serum IFN-α concentrations and (D) tissue and serum viral titers (the x-axis represents the limit of detection) are shown. \*\*p=0.0060, \*\*\*\*p<0.0001 compared to sham-infected normal controls. SI, sham-infected.

The online version of this article includes the following source data for figure 4:

**Source data 1.**

of hTfR1 HOM mice with JUNV resulted in significant clinical disease, which progressed to uniform lethality within 2 weeks of infection (*Figure 7A–C*). In contrast, illness was not observed in hTfR1 HOM–IFN-α/β receptor (R)-deficient or hTfR1 HOM–IFN-α/β and -γR-deficient mice. At day 28 p.i., serum and tissue were collected from the surviving animals and titrated for infectious viral loads. JUNV was undetectable in WT mice and was present only in brain tissue of IFN-α/βR-deficient mice (*Figure 7D*). In contrast, IFN-α/β and -γR-deficient mice, and both IFN-α/βR and IFN-α/β and -γR-deficient mice expressing hTfR1, were harboring JUNV in most tissues.

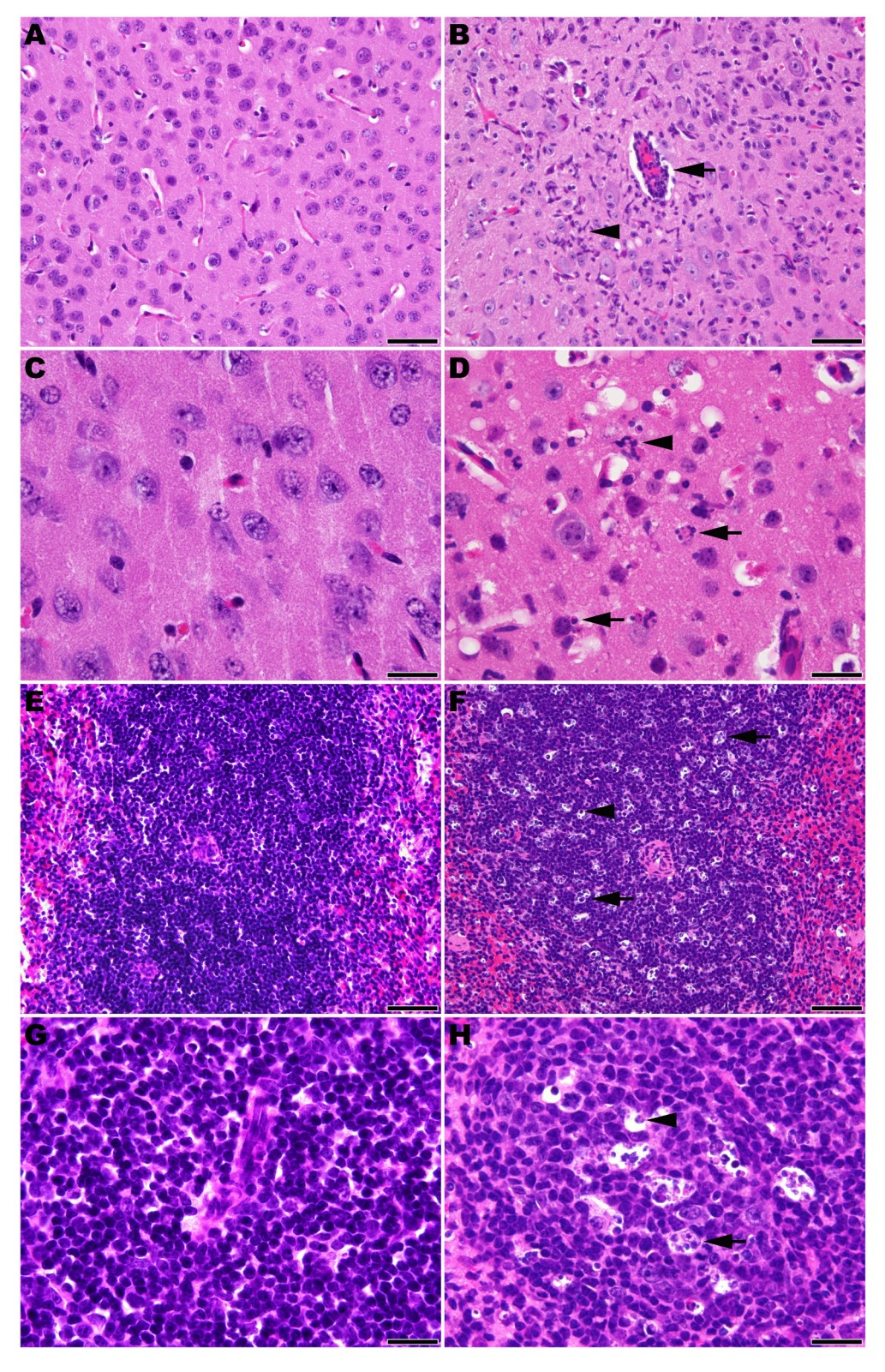

**Figure 5.** Histopathology in 3-week-old hTfR1 HOM mice infected with JUNV. Representative sections of brain (cerebral cortex) from (**A** and **C**) a sham-infected mouse and (**B** and **D**) a JUNV-infected mouse at day 12 p.i. (**B**) Neutrophilic encephalitis characterized by a perivascular cuff of inflammatory cells (arrow) and neutrophilic infiltration in the neuropil (arrowhead). (**D**) Higher magnification image showing necrotic cells (arrows) and neutrophils (arrowhead) within the neuropil. (**E** and **G**) Spleen (white pulp) from a sham-infected control mouse. (**F** and **H**) Spleen (white pulp) of a JUNV-infected

*Figure 5 continued on next page*

Figure 5 continued

mouse at day 12 p.i. (F) Tingible body macrophages (arrow) and individual lymphocyte cell death (arrowhead) are scattered within a periarteriolar lymphoid sheath. (H) Higher magnification image showing tingible body macrophages with cytoplasmic engulfed apoptotic debris (arrow) and individual lymphocyte cell death (arrowhead). Hematoxylin and eosin stain. A, B, E and F: 400 × magnification, bar = 50 μm. C, D, G and H: 1000 × magnification, bar = 20 μm.

Because of the hybrid background of the genotypic variants of mice used in the initial experiment to assess the role of the type I IFN response in JUNV pathogenesis, we could not rule out the contribution of genetic variation to susceptibility to JUNV infection, disease and persistence. Thus, to further investigate the role of type I IFN in the lethal disease outcome in hTfR1 HOM mice, JUNV infection in the presence of temporary antibody-mediated blockade of type I IFN receptors was pursued. Administration of anti-IFN-α/βR monoclonal antibodies (mAbs) as a single dose the day prior to JUNV challenge, with or without additional maintenance dosing to continue to suppress the type I IFN response, was sufficient to protect hTfR1 HOM mice from lethal disease (*Figure 8A*). Notably, one of the mAb-treated mice in the one-week treatment group was euthanized on day 13 p.i. due to the development of recurring seizures of unknown etiology. While the vehicle placebo-treated animals displayed notable weight loss (>10%) and other disease signs starting on day 9 p.i., clinical disease was not observed in any of the anti-IFN-α/βR mAb-treated mice (*Figure 8B and C*). To a lesser extent than observed in the previous experiment in hTfR1 HOM mice lacking type I and type II IFN receptors, JUNV persisted in apparently healthy mAb-treated mice at the conclusion of the study (*Figure 8D*). Collectively, the findings from these two experiments indicate that the type I IFN response plays a central role in the development of severe disease associated with JUNV infection in mice expressing hTfR1.

## Discussion

The first step in the viral infection process is the interaction between the virion and its host cell receptor. Pathogenic NWMs infect cells of their natural rodent reservoir via the endogenous TfR1 (*Zong et al., 2014*; *Radoshitzky et al., 2008*). These viruses have, in addition, adapted to utilize hTfR1 as the principal entry receptor in human cells (*Radoshitzky et al., 2007*). Related NWMs that enter through TfR1-independent mechanisms do not cause disease in humans (*Sarute and Ross, 2017*). Moreover, species known to be susceptible to disease when challenged with pathogenic NWMs (guinea pigs, marmosets, macaques) express TfR1 orthologs capable of mediating attachment and cellular entry (*Helguera et al., 2012*; *Hickerson et al., 2020*). By contrast, species that are refractory to pathogenic NWM infection (laboratory mice and hamsters) express TfR1 orthologs that do not support virus entry (*Hickerson et al., 2020*; *Radoshitzky et al., 2008*). Taken together, these findings suggest that the ability of NWMs to use TfR1 for entry is an important determinant in defining species vulnerability to severe disease (*Choe et al., 2011*). In the present study, we tested the hypothesis that mice genetically engineered to express hTfR1 would be susceptible to disease following exposure to JUNV. While hTfR1 HET mice were more susceptible to moderate JUNV disease than their WT counterparts, the development of uniformly lethal disease required the homozygous expression of hTfR1. The more severe disease phenotype was likely due to higher levels of hTfR1 in the liver and other tissues in the hTfR1 HOM mice compared to hTfR1 HET littermates (*Yu et al., 2014*). The susceptibility of hTfR1 HOM mice to lethal JUNV infection was age-dependent as the animals developed resistance as they matured. Several mechanisms may be involved in the development of resilience to JUNV challenge including a reduction in target receptor expression levels as mice age (*Hofer et al., 2008*; *Domellöf et al., 2014*; *Lönnerdal and Kelleher, 2007*; *Sciot et al., 1990*) and the maturation of the immune response (*Coulombié et al., 1986*; *Blejer et al., 1987*; *Blejer et al., 1986*; *Bruyns et al., 1976*; *Doria et al., 1978*; *D'Eustachio and Edelman, 1975*; *Landreth, 2002*). Additional studies are needed to investigate the age-dependent susceptibility of hTfR1 mice to JUNV and other New World hemorrhagic fever mammarenaviruses.

The type I IFN response plays an important role in protecting the host against viral pathogens; however, recent studies in mice characterizing the pathogenesis of acute infection by lymphocytic choriomeningitis virus (LCMV), a related mammarenavirus, have shown that the response can also contribute to lethal disease (*Oldstone et al., 2018*; *Baccala et al., 2014*). In the natural history and

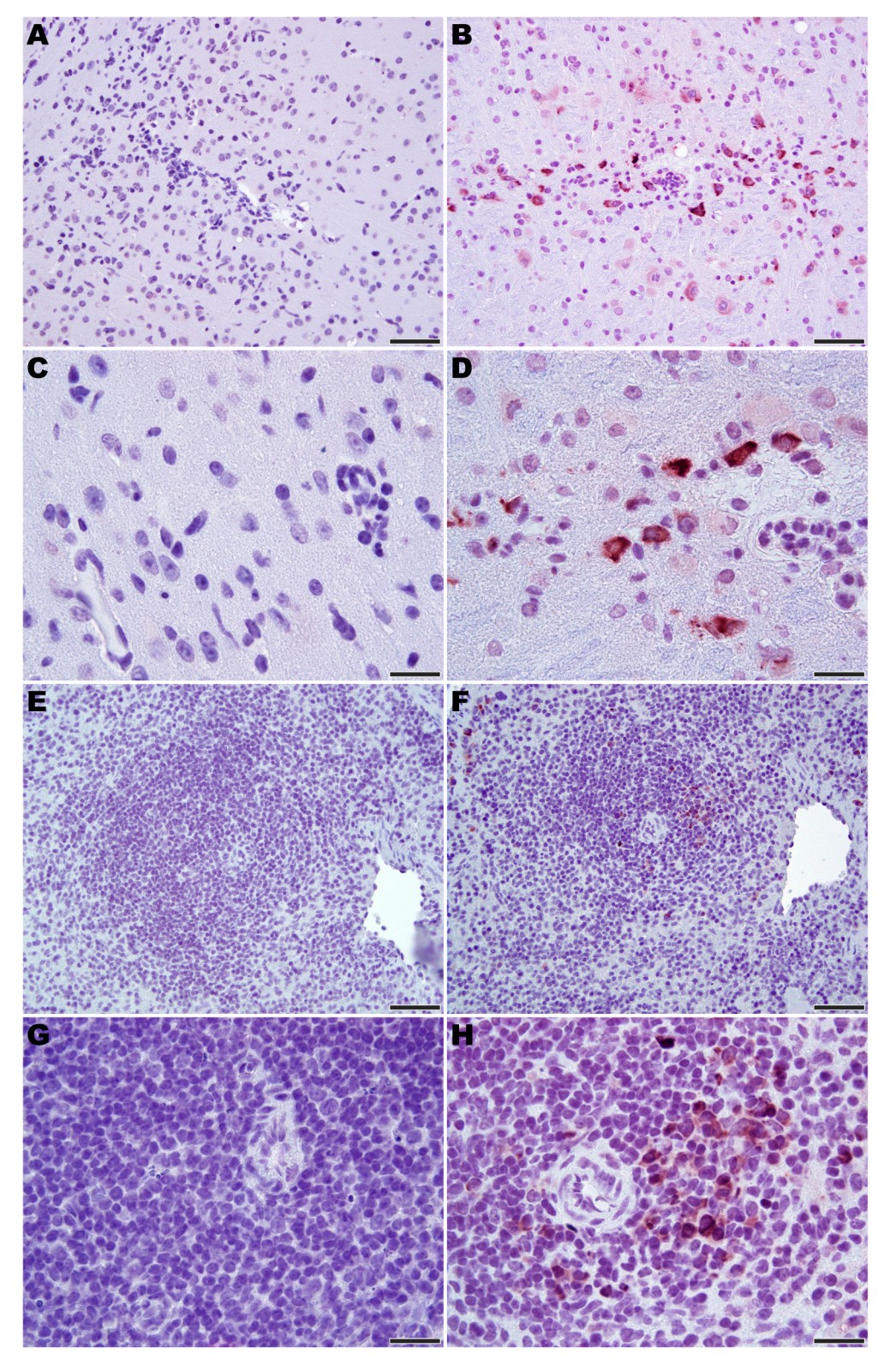

**Figure 6.** Immunohistochemistry for JUNV antigen in 3-week-old hTfR1 HOM mice. Representative sections of brain (midbrain) from (**A** and **C**) a sham-infected mouse and (**B** and **D**) a JUNV-infected mouse at day 12 p.i. Note the presence of the virus in neurons (brown staining). (**E** and **G**) Spleen (white pulp) from a sham-infected control mouse. (**F** and **H**) Virus antigen in mononuclear cells in the spleen (white pulp) of a JUNV-infected mouse at day 12 p.i. Hematoxylin counterstain. **A, B, E** and **F**: 400 × magnification, bar = 50 μm. **C, D, G** and **H**: 1000 × magnification, bar = 20 μm.

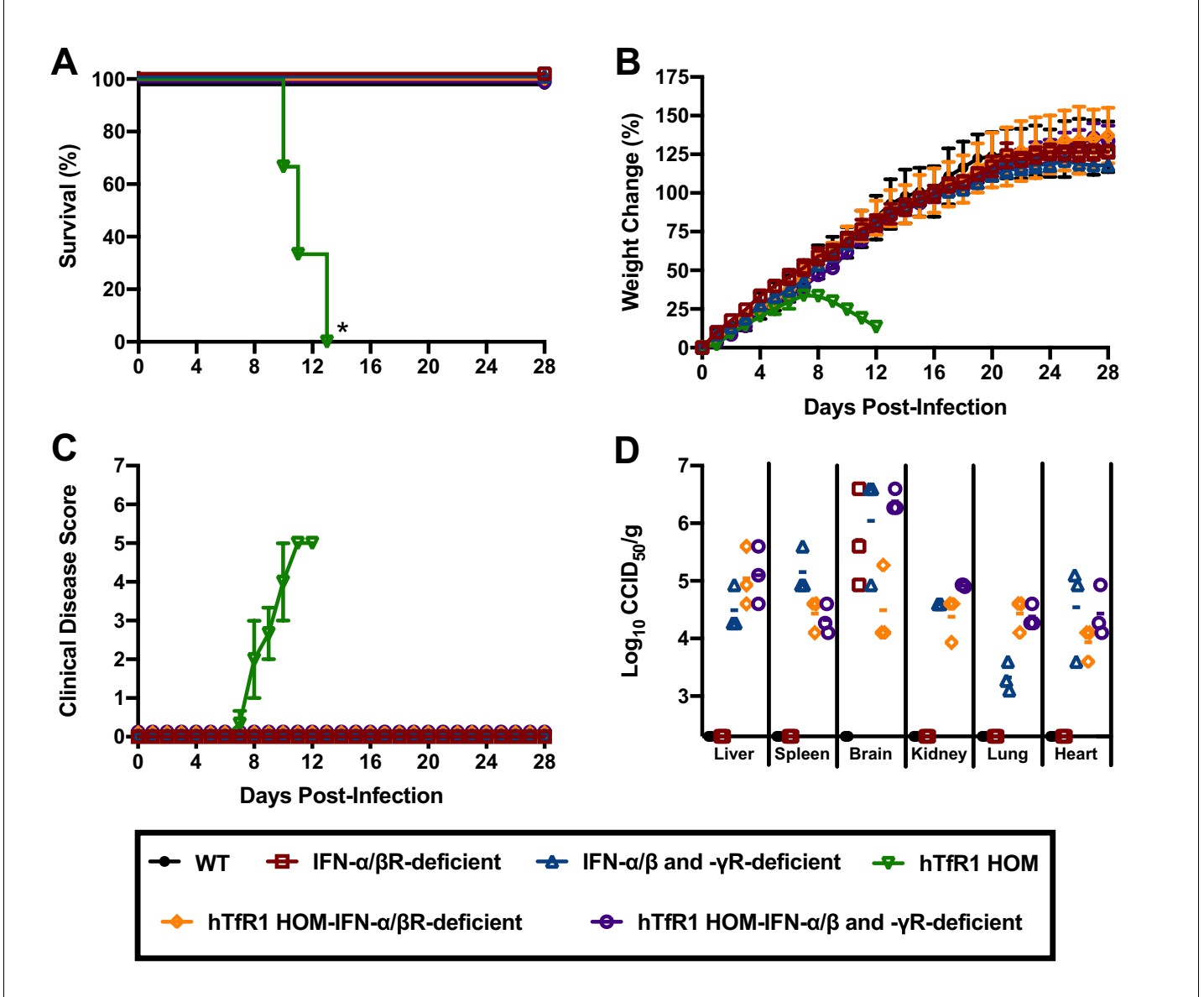

**Figure 7.** Effect of type I and II IFN response on JUNV infection and disease outcome in 3-week-old hTfR1 HOM mice. Mice (*n* = 3/group) of different phenotypic backgrounds were challenged with $10^5$ $CCID_{50}$ of JUNV and monitored daily for (**A**) survival, (**B**) weight change relative to the day of virus challenge (group mean and SEM), (**C**) clinical disease (group mean and SEM) and (**D**) tissue viral loads present on day 28 p.i. in surviving animals (virus was not detected in serum or intestine or in any WT mice). The x-axis represents the virus titer assay limit of detection. *p=0.0246 compared to WT mice or all other phenotypes.

The online version of this article includes the following source data for figure 7:

**Source data 1.**

pathogenesis study characterizing JUNV infection and disease in hTfR1 HOM mice, we found that serum levels of IFN-α began to spike on day 8 p.i., just before the onset of clinical disease signs. Notably, patients suffering from AHF have been found to have increased concentrations of circulating IFN-α, which is believed to contribute to the clinical disease manifestations (*Levis et al., 1984*; *Levis et al., 1985*). The type I IFN response (IFN-α and IFN-β), which is a critical element of the innate immune response to throttle viral infections prior to the development of the adaptive immune response, can also have a damaging effect on the host when elevated systemic levels of type I IFN persist (*Lee and Ashkar, 2018*; *Sen, 2001*). Type I IFN signaling can impair immune system function by enhancing the expression of immunosuppressive molecules and inducing immune cell apoptosis

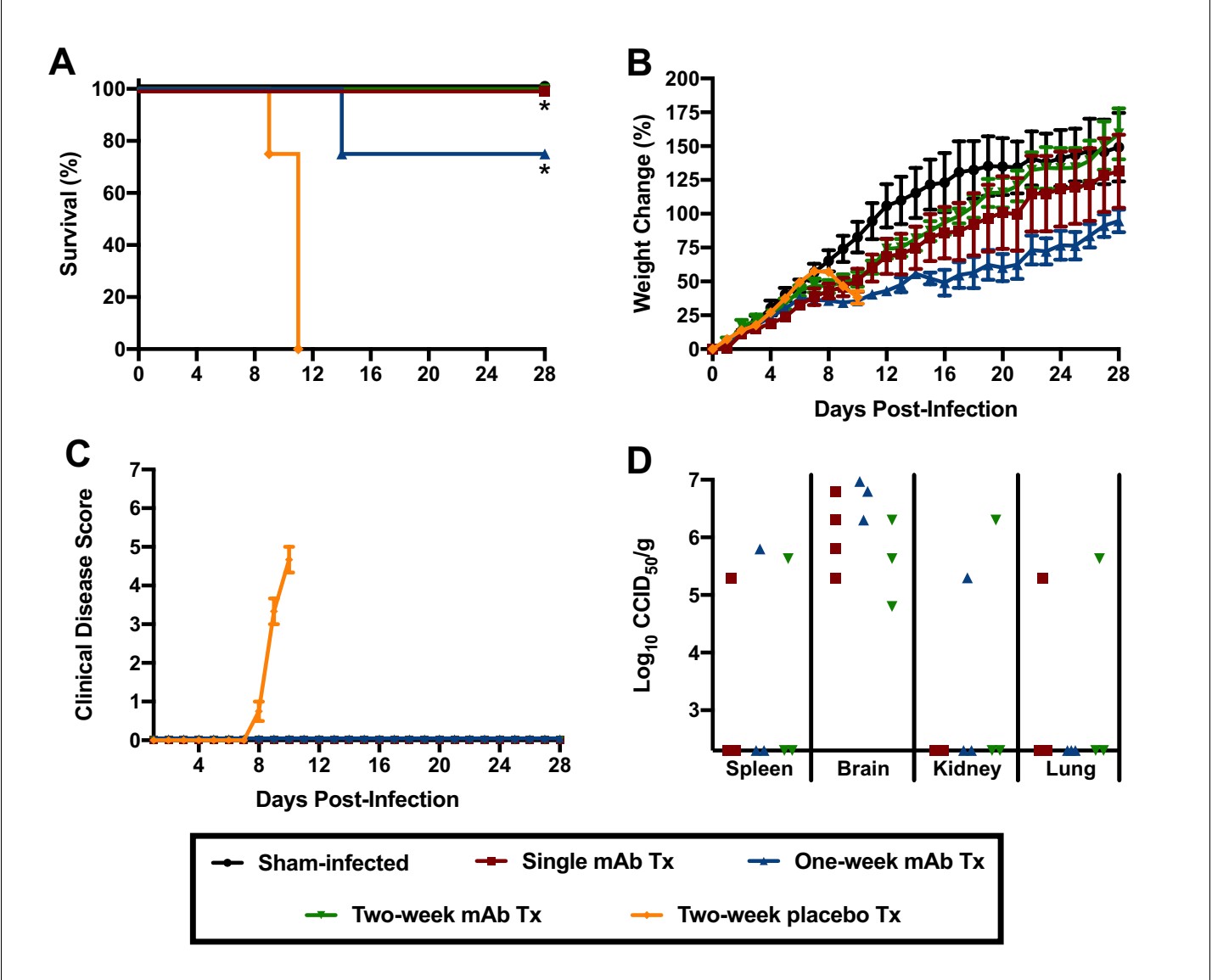

**Figure 8.** Effect of anti-IFN-α/βR mAb treatment on JUNV pathogenicity in 3-week-old hTfR1 HOM mice. Animals ($n$ = 3–4/group) were treated with a single 500 µg dose of IFN-α/βR-blocking mAb (with or without additional 250 µg maintenance doses every other day for one or two weeks) and infected i.p. the following day with $10^5$ CCID$_{50}$ of JUNV. The mice were monitored daily for (A) survival, (B) weight change relative to the day of virus challenge (mean and SEM), (C) clinical disease (mean and SEM) and (D) tissue viral loads in surviving animals on day 28 p.i. (virus was undetectable in serum, liver, lung, heart and intestine). The x-axis represents the virus titer assay limit of detection. *p=0.0114 comparing the two-week placebo treatment to the single and one-week mAb treatments; *p=0.0238 comparing the two-week placebo treatment to the two-week mAb treatment. Tx, treatment.

The online version of this article includes the following source data for figure 8:

**Source data 1.**

(*Stifter and Feng, 2015*), which has been shown to affect the development of an effective adaptive immune response and contributes to the inability of the host to resolve persistent LCMV infections (*Teijaro et al., 2013*; *Wilson et al., 2013*). In lethal NZB and FVB/N mouse infection models based on intravenous challenge with 2 × $10^6$ plaque-forming units of the Clone 13 (Cl13) LCMV variant, type I IFN was found to underlie the severe disease phenotype (*Oldstone et al., 2018*; *Baccala et al., 2014*). Similarly, in hTfR1 HOM mice lacking IFN-α/βR or IFN-α/β and -γ receptors, or hTfR1 HOM mice treated with mAbs that block the IFN-α/βR, we demonstrate the essential role of the type I IFN response in driving JUNV pathogenesis. Deletion of the CD8 T cell response also

prevents lethality associated with LCMV Cl13 infection in NZB and FVB/N mice (*Oldstone et al., 2018*; *Schnell et al., 2012*). The role of cytotoxic and helper T cells in JUNV pathogenesis in hTfR1 was not defined in the present study, but comparisons to the NZB and FVB/N models are complicated by differences in mouse age and genetics, as well as the virus challenge route and infectious doses administered. With respect to the latter, the outcome of infection (persistence vs. immune-mediated lethality) with LCMV Cl13 in C57BL/6 mice could be manipulated by modifying the virus challenge dose (*Waggoner et al., 2012*; *Cornberg et al., 2013*).

Mammarenaviruses are well known for their ability to cause persistent infections in rodents. Despite the lack of observable disease in the absence of IFN-α/β or IFN-α/β and -γ receptors, or following temporary mAb blockade of the IFN-α/βR, JUNV was able to persist in hTfR1 mice for up to 28 days in multiple tissues. It is likely that in the absence of type I IFN signaling JUNV infection would persist for an extended period (60 days), similar to that observed in Cl13 LCMV-infected mice protected from lethal disease through mAb treatment to block IFN-α/βR signaling (*Oldstone et al., 2018*). Also of note, JUNV persisted in multiple tissues of IFN-α/β and -γR-deficient mice expressing native mouse TfR1, while viral persistence in IFN-α/βR-deficient mice expressing mouse TfR1 was only evident in the brain. Consistent with the antiviral role of IFN-γ (*Schroder et al., 2004*), the dual type I and type II IFN receptor deficiency was more permissive to persistence regardless of whether or not hTfR1 was present. Further studies investigating long-term persistence and whether JUNV is shed in fecal matter and urine are needed to shed light on the potential virus-host carrier state that may result from infection in hTfR1 mice with compromised type I IFN responses.

In humans, mammarenaviral hemorrhagic fever is generally associated with mild pathological changes (*Paessler and Walker, 2013*). JUNV infection in hTfR1 mice resulted in limited histologic lesion development observed only during advanced stages of disease. Cell death of splenic mononuclear cells was present in mice at 10 and 12 days p.i. This lesion has been observed in other animal models of JUNV infection and natural infection in humans (*Weissenbacher et al., 1975*; *Kenyon et al., 1985*; *González et al., 1980*; *Carballal et al., 1981*). In humans, JUNV has been detected in phagocytic cells but not lymphocytes (*González et al., 1980*). In hTfR1 mice, the presence of JUNV antigen in the spleen was detected mainly in the white pulp suggesting a cytopathic effect on mononuclear cells. Similar findings have also been observed in the guinea pig model for JUNV (*Yun et al., 2008*), which suggests that targeting of mononuclear phagocytes may be central to pathogenesis. Death of the hTfR1 mice was tentatively attributed to neutrophilic encephalitis with JUNV antigen detected in neurons. Encephalitis has been reported in primates and humans infected with JUNV (*González et al., 1983*; *Weissenbacher et al., 1979*). Increased vascular permeability and prominent histopathology are features observed with LCMV Cl13 infection in NZB and FVB/N mice that succumb from respiratory failure and shock (*Oldstone et al., 2018*; *Baccala et al., 2014*; *Schnell et al., 2012*; *Puglielli et al., 1999*). Additional studies are needed to determine whether vascular leak associated with excess and prolonged proinflammatory cytokine concentrations, including IFN-α, may also be contributing to the demise of the JUNV-infected hTfR1 HOM mice.

The use of hTfR1 as the primary cellular entry receptor renders the pathogenic NWMs vulnerable to the development of countermeasures that interfere with the interaction between the viral envelope glycoprotein GP1 attachment subunit and the host cell receptor. The region of hTfR1 that mediates binding to the ectodomain of GP1 is the apical domain, which is not involved in binding to the known principal ligands, transferrin and hemochromatosis protein (*Cheng et al., 2004*; *Lawrence et al., 1999*; *Montemiglio et al., 2019*), thus representing a potential host-directed therapeutic target to broadly inhibit infection by JUNV and other New World hemorrhagic fever mammarenaviruses. Recent advances to exploit this vulnerability include the identification of a mAb (ch128.1) and an aptamer that bind to the apical domain of hTfR1 and the development of an immunoadhesin with the sequence of white-throated woodrat TfR1 apical domain fused to the Fc region of an IgG which binds to pathogenic NWM GP1 (*Helguera et al., 2012*; *Maier et al., 2016*; *Cohen-Dvashi et al., 2020*). Until now, proof-of-concept experiments to determine whether the ch128.1 strategy would prove to be an effective host-directed therapeutic to broadly treat NWM hemorrhagic fever were not possible due to the lack of an appropriate small-animal model. The novel hTfR1 mouse JUNV infection model that we have developed will be a valuable system in which this type of specific intervention can be investigated due to the requirement of hTfR1. Moreover, the new mouse model also provides an alternative to evaluate experimental therapies that directly target attachment by JUNV and related pathogenic NWMs and benefits from the large number of

validated mouse reagents to assess immunological parameters and other physiological processes. By comparison, studies using the guinea pig JUNV infection model are limited by a shortage of reagents, animal costs and the requirement for substantially higher quantities of investigational drugs, which is often a limiting factor during early stages of antiviral drug discovery and development.

In summary, our findings identify hTfR1-mediated entry and the type I IFN response as key factors in the development of lethal JUNV disease in mice. The development of the first mouse model of JUNV infection in immunocompetent mice will be useful for investigating JUNV pathogenesis and early preclinical development of promising therapeutic interventions including approaches that disrupt the pathogenic NWM GP1-hTfR1 apical domain interaction. In addition, the identification of the type I IFN response as a key element in the development of severe JUNV disease in hTfR1 mice further supports investigation and development of immune-modulating agents as potential therapies to limit disease severity in cases of AHF and other NWM hemorrhagic fevers.

# Materials and methods

## Key resources table

| Reagent type (species) or resource | Designation | Source or reference | Identifiers | Additional information |
|---|---|---|---|---|
| Genetic reagent (*Mus musculus*) | C57BL/6 hTfR1 knock-in mice (human *TFRC* replacing the mouse *Tfrc*) | Genentech | | |
| Genetic reagent (*Mus musculus*) | AG129 mice deficient in IFN-α/β receptor (R) and IFN-γR ($Ifnar^{-/-}$; $Ifngr^{-/-}$) | Washington University Medical School | | |
| Genetic reagent (*Mus musculus*) | Wild-type (WT) mice (hybrid C57BL/6 × 129 background) | This paper | | See Materials and methods |
| Genetic reagent (*Mus musculus*) | hTfR1 heterozygous (HET) mice (hybrid C57BL/6 × 129 background) | This paper | | See Materials and methods |
| Genetic reagent (*Mus musculus*) | hTfR1 homozygous (HOM) mice (hybrid C57BL/6 × 129 background) | This paper | | See Materials and methods |
| Genetic reagent (*Mus musculus*) | IFN-α/βR-deficient mice (hybrid C57BL/6 × 129 background) | This paper | | See Materials and methods |
| Genetic reagent (*Mus musculus*) | IFN-α/β and -γR-deficient mice (hybrid C57BL/6 × 129 background) | This paper | | See Materials and methods |
| Genetic reagent (*Mus musculus*) | hTfR1 HOM–IFN-α/βR-deficient mice (hybrid C57BL/6 × 129 background) | This paper | | See Materials and methods |
| Genetic reagent (*Mus musculus*) | hTfR1 HOM–IFN-α/β and -γR-deficient mice (hybrid C57BL/6 × 129 background) | This paper | | See Materials and methods |
| Strain, strain background (Junín virus) | Recombinant JUNV Romero strain | University of Texas Medical Branch | | |
| Cell line (*Cercopithecus aethiops*) | Vero | ATCC | Cat# CCL-81 RRID:CVCL_0059 | |

*Continued on next page*

*Continued*

| Reagent type (species) or resource | Designation | Source or reference | Identifiers | Additional information |
|---|---|---|---|---|
| Commercial assay or kit | VeriKine Mouse Interferon Alpha ELISA Kit | PBL Assay Science | Cat# 42115–1 | |
| Antibody | Goat anti-mouse IgG1 Fab (polyclonal) | Jackson ImmunoResearch Laboratories | Cat# 115-007-185 RRID:AB_2632498 | (1:10) |
| Antibody | Anti-JUNV nucleoprotein antibody (QC03-BF11) | BEI Resources | Cat# NR-43775 | (1:100) |
| Antibody | Goat anti-mouse IgG (H+L) - HRP secondary antibody (polyclonal) | Thermo Fisher Scientific | Cat# G-21040 RRID:AB_2536527 | (1:100) |
| Antibody | Anti-mouse IFNAR-1 (anti-IFN-$\alpha/\beta$R) monoclonal antibody (MAR1-5A3) | Bio X Cell | Cat# BE0241 RRID:AB_2687723 | (500 µg primary dose; 250 µg maintenance dose) |
| Software | GraphPad Prism software | GraphPad Prism (https://www.graphpad.com) | RRID_SCR_002798 | Version 8.4.1 |

## Virus and cells

The molecular clone of the Romero strain of JUNV (*Emonet et al., 2011*) was kindly provided by Dr. Slobodan Paessler (University of Texas Medical Branch, Galveston, TX). The virus stock ($10^7$ $CCID_{50}$/ml) was prepared from a single passage in Vero African green monkey kidney (ATCC CCL-81) cells (American Type Culture Collection, Manassas, VA) maintained in minimal essential medium (MEM) supplemented with 10% fetal bovine serum (HyClone, Logan, UT). Low-passage cells grown directly from the Vero stock generated from the first passage of the CCL-81 source vial obtained from ATCC were used for all experiments. The Vero cell stock was confirmed to be free of mycoplasma using the PlasmoTest – Mycoplasma Detection Kit (InvivoGen, San Diego, CA). The virus stock was diluted in MEM vehicle to achieve the desired viral doses in a 0.1 ml volume. All work with JUNV was conducted in enhanced biosafety level 3+ containment facilities at USU by Candid#1-vaccinated personnel.

## Animals

C57BL/6 hTfR1 knock-in (human *TFRC* replacing the mouse *Tfrc*) mice were obtained from Genentech (San Francisco, California) and have been previously described (*Yu et al., 2014*). Heterozygous (HET) hTfR1 mice were bred to produce homozygous (HOM) hTfR1 founders. The founding animals were backcrossed twice with AG129 mice, a 129/SvEv strain deficient in type I and type II IFN receptors (IFN-$\alpha/\beta$ and -$\gamma$R-deficient), and the resulting hybrid animals were crossed to produce 1) WT, 2) hTfR1 HET, 3) hTfR1 HOM, 4) IFN-$\alpha/\beta$R-deficient, 5) IFN-$\alpha/\beta$ and -$\gamma$R-deficient, 6) hTfR1 HOM–IFN-$\alpha/\beta$R-deficient and 7) hTfR1 HOM–IFN-$\alpha/\beta$ and -$\gamma$R-deficient mice. The AG129 mice were kindly provided by Dr. Robert Shreiber (Washington University Medical School, St. Louis, MO). WT littermates and mice designated hTfR1 HET or hTfR1 HOM expressed both IFN-$\alpha/\beta$ and -$\gamma$ receptors. All mice were genotyped for the presence or absence of mouse TfR1 and hTfR1 and type I and type II IFN receptors by PCR. Male and female animals were used in all studies.

## Susceptibility of hTfR1 mice to lethal JUNV infection

In 2 separate experiments, hybrid 3-week-old WT, hTfR1 HET and hTfR1 HOM mice ($n$ = 8–9/virus challenge group, $n$ = 6 for the sham-infected control group which included 2 mice of each genotype) were inoculated with $10^5$ $CCID_{50}$ of JUNV or sham-infected with MEM only via 0.1 ml i.p. injection. The number of mice per group was selected based on previous experience resolving differences in survival outcomes in uncharacterized rodent models of human viral diseases. Following JUNV

challenge, the mice were weighed and observed daily for 21 days for morbidity and mortality. By 21 days p.i., survivors were generally recovering from the infection, as judged by normal activity and body condition.

## Age-dependent susceptibility of hTfR1 HOM mice to lethal JUNV infection

Cohorts of 3, 4, 5 and 6-week-old hTfR1 HOM mice ($n = 6$/JUNV infection group for the 3- and 4-week-old mice, $n = 3$/JUNV infection group for the 5- and 6-week-old mice and $n = 3$/age-matched sham-infected controls per age group) were challenged i.p. with $10^5$ CCID$_{50}$ of JUNV or sham-infected with MEM vehicle. Following challenge, the mice were weighed and monitored daily for morbidity and mortality for 28 days. For this and all subsequent experiments, clinical signs of disease were scored as 0 (not present) or 1 (present) based on the presence of the following disease signs: weight loss exceeding 10% of peak weight, lethargy, hunched posture, ruffled fur, tremors, paralysis, distended abdomen and bleeding. Animals with a cumulative clinical score greater than 6, experiencing weight loss greater than 30% compared to peak weight or unresponsive to external stimulus, were euthanized.

## Lethal dose determination in hTfR1 HOM mice

To determine the LD$_{50}$ and LD$_{90}$ of JUNV in hTfR1 HOM mice, groups of 3-week-old mice ($n = 7$/virus challenge dose, $n = 3$/sham-infected control group) were inoculated by i.p. injection with one of three serial log$_{10}$ dilutions ($10^5$, $10^4$ or $10^3$ CCID$_{50}$) of JUNV or sham-infected with MEM. Mice were weighed the day before JUNV challenge and assigned to experimental groups to minimize sex and weight differences across the groups. For 28 days, the animals were weighed daily and assigned a score from 0 to 8 based on clinical signs of disease. The LD$_{50}$ and LD$_{90}$ values were calculated using Prism 8 (version 8.4.1; GraphPad, La Jolla, CA).

## Natural history and pathogenesis of JUNV infection in hTfR1 HOM mice

Groups of 3-week-old hTfR1 mice were challenged with $10^4$ CCID$_{50}$ of JUNV or sham-infected. Mice were weighed the day before virus challenge and assigned to experimental groups to minimize sex and weight differences across the groups. Cohorts of 4 animals/group were euthanized every other day beginning on day 2 p.i. No animals survived beyond 12 days p.i. A single sham-infected control mouse was euthanized each day on days 2, 6 and 10 p.i. During the course of the experiment and prior to euthanasia, mice were weighed and evaluated daily for clinical disease signs. Blood samples were collected by submandibular vein puncture to obtain serum for analysis of IFN-$\alpha$ concentration and viremia. Following euthanasia, mice were transcardially perfused with sterile phosphate-buffered saline (PBS) before tissue samples of brain, liver, spleen, lung, heart, kidney and intestine were collected for determination of viral load and histopathology.

## Determination of tissue and serum viral loads

Viral loads in tissues and serum were assayed using an infectious cell culture assay, as previously described (Gowen et al., 2007). Briefly, tissues were homogenized in a fixed volume of MEM and the homogenates and serum were serially diluted and added to quadruplicate wells of Vero cell monolayers in 96-well microtiter plates. Viral cytopathic effect was determined 10 days p.i., and the 50% endpoints were calculated by the Reed and Muench method (Reed, 1938). The assay limits of detection were 1.67 log$_{10}$ CCID$_{50}$ per ml of serum and 2.23 log$_{10}$ CCID$_{50}$/g of tissue.

## Serum IFN-$\alpha$ analysis

Serum IFN-$\alpha$ concentration for each mouse was determined using the VeriKine Mouse Interferon Alpha ELISA Kit (PBL Assay Science, Piscataway, NJ). The assay was completed following the manufacturer's specifications.

## Histopathology and immunohistochemistry (IHC)

Samples of brain, liver, spleen, lung, heart, kidney and intestine of hTfR1 mice collected at 2, 4, 6, 8, 10 and 12 days p.i. were preserved in 10% neutral buffered formalin. Fixed tissue samples were processed and embedded in paraffin according to routine histologic techniques. Tissue sections, 5

μm thick, were stained with hematoxylin and eosin and examined by light microscopy by a board-certified pathologist who was blinded to the groups and day of euthanasia. Tissue lesions severity were scored as follows: 0 = no lesions, 1 = minimal, 2 = mild, 3 = moderate and 4 = severe. Formalin-fixed sections of spleen, liver, kidney, intestine, lung and brain from sham-infected and moribund JUNV-infected hTfR1 HOM mice at day 12 p.i. were also evaluated for the presence of JUNV antigen by IHC using a mouse mAb raised against the JUNV nucleoprotein (BEI Resources, Manassas, VA). Briefly, tissue sections were permeabilized with 0.5% Triton X-100 (Sigma-Aldrich, St. Louis, MO) for 5 min and the endogenous peroxidase activity was blocked for 15 min with 3% hydrogen peroxide. Prior to incubation with the primary anti-JUNV nucleoprotein mAb, the slides were incubated with blocking solution (PBS containing 10% normal goat serum and 0.2% Triton X 100) for 1 hr. To block background staining from endogenous mouse IgG, the tissue sections were incubated with goat anti-mouse IgG1 Fab (1:10 dilution; Jackson ImmunoResearch, West Grove, PA) for 1 hr. After blocking, the slides were incubated with mouse anti-JUNV nucleoprotein antibody, diluted 1:100 in blocking solution, for 24 hr. Finally, the slides were incubated for 1 hr with goat anti-mouse IgG (H+L)-HRP secondary antibody (1:100 dilution; Thermo Fisher Scientific, Waltham, MA) and developed using ImmPACT NovaRED Peroxidase Substrate (Vector Laboratories, Burlingame, CA) according to the manufacturer's specifications. The slides were counterstained with hematoxylin.

### Susceptibility of hTfR1 HOM mice lacking type I IFN receptor, or both type I and type II IFN receptors, to JUNV infection

Cohorts of 3-week-old mice ($n$ = 3/group) representing 6 different genetic profiles (WT, IFN-$\alpha/\beta$R-deficient, IFN-$\alpha/\beta$ and -$\gamma$R-deficient, hTfR1 HOM, hTfR1 HOM–IFN-$\alpha/\beta$R-deficient and hTfR1 HOM–IFN-$\alpha/\beta$ and -$\gamma$R-deficient) were challenged i.p. with $10^5$ CCID$_{50}$ JUNV. The number of mice per group was based on power analysis performed using commonly accepted values for type I error (0.05) and power (80%). After challenge, the animals were weighed and assigned a clinical score daily for 28 days. At the end of the study, serum and tissues were collected from the surviving animals and assessed for viral burden.

### Blockade of the type I IFN receptor in hTfR1 HOM mice and susceptibility to JUNV infection

Three-week-old hTfR1 HOM mice were weighed the day before JUNV challenge and assigned to experimental groups ($n$ = 3–4/virus challenge group, $n$ = 3 for the sham-infected group) to minimize sex and weight differences across the groups. The number of mice per group was based on power analysis performed using commonly accepted values for type I error (0.05) and power (80%). The mice were administered anti-IFN-$\alpha/\beta$R mAbs (500 μg; MAR1-5A3; Bio X Cell, West Lebanon, NH) or placebo (PBS vehicle only) via i.p. injection, 24 hr before i.p. challenge with $10^5$ CCID$_{50}$ of JUNV. Following infection, specified groups of mice received additional i.p. injections of 250 μg of the anti-IFN-$\alpha/\beta$R mAb or placebo every other day for 1 or 2 weeks. The animals were weighed and scored for clinical disease presentation daily for 28 days. At the conclusion of the study, serum and tissues were harvested from the surviving animals and viral loads determined.

### Statistical analysis

The log-rank test was used for the analysis of Kaplan-Meier survival curves. A one-way analysis of variance (ANOVA) with Dunnett's multiple comparisons test was performed to compare differences in serum IFN-$\alpha$ concentrations. Weight change curves of groups where all animals survived were compared by two-way repeated-measures ANOVA with Sidak multiple comparisons test. All statistical evaluations were performed using Prism 8 (version 8.4.1). Results were considered significant if p$\leq$0.05.

## Additional information

### Competing interests
Manuel L Penichet: is a shareholder of Klyss Biotech, Inc. The Regents of the University of California licensed Dr. Penichet's technology to this firm. The author has no other competing interests to declare. The other authors declare that no competing interests exist.

### Funding

| Funder | Grant reference number | Author |
| --- | --- | --- |
| National Institutes of Health | R56 AI13646 | Brian B Gowen |
| National Institutes of Health | R01 AI14110 | Brian B Gowen |
| National Institutes of Health | R01 CA196266 | Manuel L Penichet |
| Utah State University | Graduate Research and Creative Opportunities grant | Brady T Hickerson |

The funders had no role in study design, data collection and interpretation or the decision to submit the work for publication.

### Author contributions
Brady T Hickerson, Conceptualization, Data curation, Formal analysis, Funding acquisition, Investigation, Methodology, Writing - original draft, Writing - review and editing; Eric J Sefing, Kevin W Bailey, Investigation; Arnaud J Van Wettere, Formal analysis, Visualization, Methodology, Writing - review and editing; Manuel L Penichet, Conceptualization, Resources, Formal analysis, Funding acquisition, Investigation, Methodology, Writing - review and editing; Brian B Gowen, Conceptualization, Data curation, Formal analysis, Supervision, Funding acquisition, Investigation, Methodology, Writing - original draft, Project administration, Writing - review and editing

### Author ORCIDs
Brian B Gowen https://orcid.org/0000-0001-9113-2575

### Ethics
Animal experimentation: All animal procedures complied with USDA guidelines and were conducted at the AAALAC-accredited laboratory animal research facilities at Utah State University under protocol #10034, approved by the Utah State University Institutional Animal Care and Use Committee.

### Decision letter and Author response
Decision letter https://doi.org/10.7554/eLife.55352.sa1
Author response https://doi.org/10.7554/eLife.55352.sa2

## Additional files

### Supplementary files
• Transparent reporting form

### Data availability
All data generated or analyzed during this study are included in the manuscript.

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
