## [Decision Letter]

**Acceptance summary:**

Current animal models of arenavirus pathogenesis are either not particularly robust or lack broad tractability. This study establishes a new mouse model of lethal Junin virus infection that recapitulates many key features of Junin virus pathogenesis in humans and may have broad utility for the development of vaccines and therapeutics targeting this important human pathogen.

**Decision letter after peer review:**

Thank you for submitting your article "Type I interferon underlies severe disease associated with Junin virus infection in mice" for consideration by *eLife*. Your article has been reviewed by three peer reviewers, one of whom is a member of our Board of Reviewing Editors, and the evaluation has been overseen by Karla Kirkegaard as the Senior Editor. The following individuals involved in review of your submission have agreed to reveal their identity: Juan Carlos de la Torre (Reviewer #2).

The reviewers have discussed the reviews with one another and the Reviewing Editor has drafted this decision to help you prepare a revised submission.

Summary:

The New World mammarenavirus Junin (JUNV) is the causative agent of Argentine hemorrhagic fever (AHF), a disease endemic to the Pampas region of Argentina that is associated with severe clinical symptoms including hemorrhagic and neurological manifestations and a case fatality rate of 15-30% in patients who do not receive appropriate supportive treatment. In addition, JUNV possesses features that make it a credible biodefense threat. Therefore, the importance of developing vaccines and therapeutics to counteract JUNV infections and associated disease. The availability of a mouse model of JUNV infection would facilitate the development of therapeutics and vaccines against AHF disease, as well as a better understanding of the molecular and cellular bases underlying JUNV pathogenesis.

Here, the authors use transgenic mice expressing TfnR, a Junin virus entry factor, to establish an immunocompetent mouse model of Junin virus infection that recapitulates several disease outcomes observed in humans. Lethal disease in infected mouse was associated with increased type I interferon responses, demonstrating a key pathological feature of human disease in these mice. The pathogenic effects could be mitigated in mice lacking IFN receptors, or in mice treated with IFN receptor blocking antibody.

Essential revisions:

1) The authors state that type I IFN underlies disease in this mouse model, but this is only demonstrated in young mice. Was viremia and serum IFN levels ever examined in older mice? Are these older mice just not replicating the virus, or is the virus replicating, but the mice are not making IFN? If the latter, would recombinant IFN treatment + Junin virus synergize in older mice to give a lethal phenotype?

2) Tissue viral loads shown in Figure 7D and 8D provide no information regarding the contribution of viral loads to pathogenesis, because they were measured at the late time period and because controls are missing. To be meaningful, viral loads should be measured at 11 or 12 dpi including control groups: hTfR1+/+ (7D) and "Placebo Tx" (8D) groups. Related to this issue, the authors stated "in the absence of a type I IFN response during the acute JUNV infection period hTfR1+/+ mice failed to clear the virus despite no apparent signs of illness". For this to be true, viral loads should be compared to those of control groups. In addition, 6 or 3-4 mice in Figure 7 and 8, respectively, may provide statistically significant survival data, but when population variance is huge like the viral loads in Figure 7D and 8D, 3-6 mice is not nearly sufficient. In order to use right sample size, power and sample size calculation should be redone based on the preliminary data in 7D and 8D.

3) Although hTfR1+/+ mice and IFNa/bR-/- and IFNa/bgR-/- mice are in different genetic background, they were backcrossed only twice. The authors' conclusion of type I IFN response being the key element for JUNV pathogenesis is partially based on the data obtained using these mice with mosaic genetic background (Figure 7). Combined with the data in Figure 8, this conclusion may still be valid, but nonetheless the authors should indicate contaminating 129 genes could have contributed to the susceptibility to viral infection and tissue viral loads.

[Editors' note: further revisions were suggested prior to acceptance, as described below.]

Thank you for resubmitting your article "Type I interferon underlies severe disease associated with Junín virus infection in mice" for consideration by *eLife*. Your article has been reviewed by two peer reviewers, and the evaluation has been overseen by a Reviewing Editor and Karla Kirkegaard as the Senior Editor.

The reviewers have discussed the reviews with one another and the Reviewing Editor has drafted this decision to help you prepare a revised submission.

Summary:

The responses provided by the authors are overall reasonable, and they have incorporated changes to the text of the revised version of the paper that address many, but not all, the criticisms raised during the review of the originally submitted paper.

Revisions:

In lieu of additional experiments to address outstanding questions, the reviewers request that the authors expand the Discussion along the following lines:

A general suggestion is for the authors to add some useful topic sentences to the Discussion and, most importantly, conclude with a discussion of any other viral infections for which type 1 interferons contribute to pathogenesis.

More specifically, the authors should provide a more elaborated comparison between the mouse model of lethal JUNV infection and the NZB and FVB mouse models of LCMV infection where type I interferon was shown to be a critical factor for fatal outcome of infection.

---

## [Author Response]

Essential revisions:1) The authors state that type I IFN underlies disease in this mouse model, but this is only demonstrated in young mice. Was viremia and serum IFN levels ever examined in older mice? Are these older mice just not replicating the virus, or is the virus replicating, but the mice are not making IFN? If the latter, would recombinant IFN treatment + Junin virus synergize in older mice to give a lethal phenotype?

The questions posed by the reviewers are of great interest and will pursued in future studies. Because our focus was to develop and characterize a lethal disease model of JUNV infection in hTfR1 mice, we did not characterize the infection in older age groups of mice that were refractory to disease. With that said, the weight loss observed in the 5-week-old hTfR1 mice challenged with JUNV suggests that there was viral replication in the older mice, which would most likely be accompanied by type I IFN induction. Studies by Ross and coworkers (Lavanya et al., Sci Transl Med, 2013; Cuevas and Ross, J Virol, 2014) have shown that the attenuated JUNV Candid#1 vaccine strain can replicate in 4 and 8 to 10-week-old mice with spleen infectious virus titers present on days 8 and 7 p.i., respectively, suggesting that the more virulent JUNV Romero strain used in our studies would likely replicate to higher titers and for a longer period of time in older mice expressing hTfR1.

2) Tissue viral loads shown in Figure 7D and 8D provide no information regarding the contribution of viral loads to pathogenesis, because they were measured at the late time period and because controls are missing. To be meaningful, viral loads should be measured at 11 or 12 dpi including control groups: hTfR1+/+ (7D) and "Placebo Tx" (8D) groups.

Ideally, we would have liked to include additional animals in each group that would be euthanized just prior to when the control groups (hTfR1+/+ or Placebo Tx) would be expected to succumb for evaluation of serum and tissue viral loads. Due to our animal and mAb limitations, we were not able to add the additional viral load arms to the studies.

Although the D panels of Figures 7 and 8 do not provide information regarding the contributions of viral loads to the disease process, the data indicate that mice with compromised type I, or compromised type I and type II, IFN responses do not clear JUNV by the end of the 28-day study period. Further studies are needed to investigate long-term persistence which could provide insights into the virus-host reservoir relationship.

Related to this issue, the authors stated "in the absence of a type I IFN response during the acute JUNV infection period hTfR1+/+ mice failed to clear the virus despite no apparent signs of illness". For this to be true, viral loads should be compared to those of control groups.

This statement has been removed from the manuscript as part of the extensive revision. We now include the following statement. “Despite the lack of observable disease in the absence of IFN-α/β or IFN-α/β and γ receptors, or following temporary mAb blockade of the IFN-α/βR, JUNV was able to persist in hTfR1 mice for up to 28 days in multiple tissues.” (fourth to the last sentence of the second paragraph of the Discussion)

In addition, 6 or 3-4 mice in Figure 7 and 8, respectively, may provide statistically significant survival data, but when population variance is huge like the viral loads in Figure 7D and 8D, 3-6 mice is not nearly sufficient. In order to use right sample size, power and sample size calculation should be redone based on the preliminary data in 7D and 8D.

The purpose of the end-of-study sample collection was to qualitatively address the question of whether or not the surviving mice had cleared the JUNV infection. Future studies investigating longer-term persistence will be powered to determine statistical significance comparing titers from the different genotypes based on the findings reported in Panels 7D and 8D.

3) Although hTfR1+/+ mice and IFNa/bR-/- and IFNa/bgR-/- mice are in different genetic background, they were backcrossed only twice. The authors' conclusion of type I IFN response being the key element for JUNV pathogenesis is partially based on the data obtained using these mice with mosaic genetic background (Figure 7). Combined with the data in Figure 8, this conclusion may still be valid, but nonetheless the authors should indicate contaminating 129 genes could have contributed to the susceptibility to viral infection and tissue viral loads.

This is an excellent point. We have addressed this oversight by the addition of the following statement to the Results text. “Because of the hybrid background of the genotypic variants of mice used in the initial experiment to assess the role of the type I IFN response in JUNV pathogenesis, we could not rule out the contribution of genetic variation in susceptibility to JUNV infection, disease and persistence.” (first sentence of last paragraph of the Results section)

[Editors' note: further revisions were suggested prior to acceptance, as described below.]

Revisions:In lieu of additional experiments to address outstanding questions, the reviewers request that the authors expand the Discussion along the following lines:A general suggestion is for the authors to add some useful topic sentences to the Discussion and, most importantly, conclude with a discussion of any other viral infections for which type 1 interferons contribute to pathogenesis.More specifically, the authors should provide a more elaborated comparison between the mouse model of lethal JUNV infection and the NZB and FVB mouse models of LCMV infection where type I interferon was shown to be a critical factor for fatal outcome of infection.

As suggested, we have added useful topic sentences to the Discussion.

Importantly, we have added text to paragraphs two, three and four of the Discussion, specifically elaborating on the comparison between the mouse JUNV infection model and the NZB and FVB mouse models of LCMV infection.